# Exploring the Synergistic Effect of Short Aramid Fibers and Graphene Nanoplatelets on the Mechanical and Dynamic Mechanical Properties of Polypropylene Composites Prepared via Thin-Plate Injection

**DOI:** 10.3390/polym17030374

**Published:** 2025-01-30

**Authors:** Andressa Antunes Carneiro, Iaci Miranda Pereira, Rafael Rodrigues Dias, Dionisio da Silva Biron, Heitor Luiz Ornaghi Júnior, Francisco Maciel Monticeli, Daiane Romanzini, Ademir José Zattera

**Affiliations:** 1Postgraduate Program in Engineering of Processes and Technologies (PGEPROTEC), University of Caxias do Sul (UCS), R. Francisco Getúlio Vargas, 1130, Caxias do Sul 95070-560, RS, Brazil; aacarneiro@ucs.br (A.A.C.); birondionisio@gmail.com (D.d.S.B.); ornaghijr.heitor@gmail.com (H.L.O.J.); ademirjzattera@gmail.com (A.J.Z.); 2Centro Tecnológico do Exército (CTEx), Laboratório de Materiais, Rio de Janeiro 23020-470, RJ, Brazil; iacipere@gmail.com (I.M.P.); rrodrigues083@gmail.com (R.R.D.); 3Department of Aerospace Structures and Materials, Faculty of Aerospace Engineering, Delft University of Technology, 2629 HS Delft, The Netherlands; 4Postgraduate Program in Materials, Technology and Engineering (PPGTEM), Federal Institute of Rio Grande do Sul (IFRS), R. Princesa Isabel, 60, Feliz 95770-000, RS, Brazil; daiane.romanzini@feliz.ifrs.edu.br

**Keywords:** composites, aramid fiber, graphene, mechanical properties, thin-plate injection

## Abstract

The present study aims to evaluate thin plate-injected polypropylene (PP) composites containing short aramid fibers (AF) and graphene nanoplatelets (GNPs). The aramid fibers were manually cut to a length of 10 mm and added to the polypropylene matrix at a concentration of 10 wt.%. Additionally, GNPs were incorporated at concentrations of 0.1, 0.25, and 0.5 wt.%. Maleic anhydride grafted polypropylene (MAPP) was used at a concentration of 2 wt.% to improve the adhesion and compatibility between the polymer matrix and the fillers. Thermal analyses, tensile and flexural tests, and dynamic mechanical thermal analysis were performed, followed by statistical analysis using ANOVA and Tukey’s test. The composites demonstrated significant improvements in storage and loss moduli compared to neat polypropylene. With the addition of AF and GNPs, tensile strength increased to 46.8 MPa, which represents a 265% enhancement compared to PP. Similarly, flexural strength reached 62.4 MPa, significantly higher than the 36.73 MPa for PP, particularly for the composite containing AF and 0.25 wt.% GNPs. The results presented in this study highlight the synergistic effect of aramid fibers and GNPs on PP. These improvements make the proposed composites highly promising for a range of applications, including ballistic interlayered aramid/thin-plate laminates.

## 1. Introduction

Numerous investigations indicate that the integration of fillers, such as fibers or nanoparticles, into a polymer matrix significantly improves both mechanical and thermal characteristics, facilitating the creation of innovative materials [1,2,3,4,5,6]. The polymer matrix serves to preserve the structural integrity of the component, while the fillers contribute to the enhancement of its properties. Composite materials are particularly appealing across a range of industrial sectors, including aerospace, defense, construction, automotive, sports, and maritime industries. Additionally, to improve the compatibility between the polymer matrix and the fillers, the application of compatibilizer agents, such as maleic anhydride, has been documented. Maleic anhydride functions at the interface of organic and inorganic surfaces and is notably effective in enhancing adhesion between polar and non-polar materials [7].

Aramid fiber is renowned for its exceptional strength-to-weight ratio and is extensively used in personal protective equipment, the automotive and aerospace industries, textiles, and various military applications, including armored vehicles, vests, and helmets [8,9,10]. These properties make aramid fiber an alternative to metallic, glass, and carbon fibers in specific applications. Its inclusion enhances the effectiveness and reliability of advanced composite materials, ensuring safety in contexts that demand higher strength [11,12,13]. In a study by Ari et al. [2], the authors explored the use of aramid fibers as reinforcement in a polypropylene polymeric matrix. They found that increasing the aramid content significantly improved the mechanical properties of the composites. The effectiveness of aramid fibers was comparable to that of glass and carbon fibers, resulting in an increase in tensile strength by a factor of 1.8 compared to polypropylene [2]. In a subsequent study, Ari et al. [14] investigated the use of aramid fibers as reinforcement in other thermoplastics. They observed that incorporating aramid fibers improved the mechanical properties of polyethylene, nylon 6, and nylon 12 [14]. In another study, aramid nanofibers were dispersed in an epoxy matrix, enhancing the bonding between the crystalline aramid fibers and the epoxy matrix. Compared to neat epoxy, the epoxy modified with 0.15 wt.% aramid fibers showed increases in tensile strength, flexural strength, and impact resistance of 28.2%, 81.9%, and 81.8%, respectively [15].

On the other hand, the academic world has witnessed the rise in graphene as a revolutionary material. Its two-dimensional form, consisting of flat sheets, distinguishes it from rounded or three-dimensional structures, making it one of the most studied materials today. The graphene family includes various materials, such as reduced graphene oxide, graphene oxide, graphene nanosheets or nanoplatelets, and multilayer graphene [16]. Graphene is attracting increasing interest due to its unique mechanical, thermal, and electrical properties, making it a highly versatile and promising material. These advantageous characteristics enable applications across various sectors, from solar cells development to advanced composite materials. The broad range of potential applications for graphene has driven increasing demand for this material and its derivatives, such as graphene nanoplatelets, which consist of 3 to 10 layers of graphene. This demand is prompting the development of large-scale production methods and the enhancement of quality control techniques [17]. Graphene nanoplatelets are nanostructures composed of multiple layers of graphite, with their formation attributed to intermolecular forces known as Van der Waals forces. A review of the scientific literature shows that the addition of GNPs provides significant benefits in both the mechanical performance and thermal behavior of materials [18]. GNPs exhibit mechanical, electrical, barrier, and thermal properties similar to those of graphene, but are more cost-effective to produce [19].

Various studies highlight the synergy achieved by incorporating two or more reinforcing phases into a single continuous matrix to enhance the properties of composite materials. Papageorgiou et al. [20] investigated the effect of adding glass fibers (GF) and GNPs on the mechanical and thermal properties of polypropylene. The study revealed that these fillers had an additive effect in the composite system, resulting in a Young’s modulus up to three times greater than that of neat polypropylene. The presence of GNPs at the interface between polypropylene and GF can improve the interfacial stress transfer between the two materials. In another study [21], glass fibers coated with graphene nanoplatelets were incorporated into polypropylene at low concentrations (0.1–1 wt.%). This resulted in a 31% increase in the tensile modulus compared to the neat polymer matrix. Additionally, the tensile strength increased from 39 MPa to 45 MPa with the addition of 1 wt.% GNPs.

This research can be applied in future studies to enhance the mechanical and dynamic mechanical resistance of polyolefins, for example, in ballistic applications. For instance, the research group studied the performance of multilaminar aramid fabric panels with polyethylene films reinforced with nanoparticles [22,23,24]. The novelty of this study lies in replacing the films with thin-injected plates containing aramid and graphene nanoplatelets. Additionally, the injection method used allows for the application of post-use aramid fibers (through the use of short aramid fiber). Given this context, this preliminary study aims to evaluate the mechanical and dynamic mechanical properties of composites containing short aramid fibers and graphene nanoplatelets in a polypropylene polymer matrix. The aramid fibers were manually cut to a length of 10 mm, while the graphene nanoplatelets were added at varying concentrations (0.1, 0.25, 0.5 wt.%). Furthermore, the influence of Polybond 3200 as a compatibilizer agent was assessed.

## 2. Materials and Methods

### 2.1. Materials

Polypropylene (PP) H301, with a density of 0.905 g·cm^−3^ and a melt flow index of 10 g·min^−1^, was kindly provided by Braskem (Triunfo-RS, Brazil). The compatibilizer agent, maleic anhydride grafted polypropylene (MAPP) Polybond 3200, was supplied by SI Group—ChemPoint, containing 1.2% maleic anhydride by weight. The graphene nanoplatelets (GNPs), UGZ 1004, with a thickness of 10 to 20 layers, were produced and supplied by UCS Graphene (Caxias do Sul-RS, Brazil). The para-aramid A314 KV K129 fabric fiber, with an aerial density of 400 g·m^−2^, was provided by Barrday Advanced Material Solutions (Cambridge, MA, USA).

### 2.2. Composition of the Prepared Samples

Table 1 presents the nomenclature and composition of the samples prepared in this study. Additionally, the effect of incorporating the compatibilizer agent MAPP was evaluated. The aramid fiber was manually cut to a length of 10 mm. The optimal aramid fiber size and content were determined based on preliminary processing tests conducted in a Brabender closed mixer.

### 2.3. Sample Processing

All samples on Table 1 were prepared according to Figure 1. Double processing was performed in a Seibt single-screw extruder, model ES 35 F-R, involving two main stages. Initially, a manual premixing of the components was performed, followed by extrusion in the single-screw extruder, segmentation of the extrudate, cooling, and milling in a knife mill. To achieve improved homogenization, the material obtained from the first stage underwent a second round of milling and fragmentation. The wiring matrix was removed, and the material was fragmented manually. The temperature profile used during processing included zones at 170, 180, and 190 °C, with a screw rotation speed of 65 rpm.

For the thin plate injection stage, an injection mold was developed, as shown in Figure 2. A sketch of the current molds was created, and the measurements of the bending mold were taken to ensure the required dimensional accuracy. A preliminary study was conducted using injection simulations to determine the minimum achievable thickness, aiming for complete mold filling. Initially, the polymer matrix (100% PP) was used to test the mold’s operation. The injection parameters used are listed in Table 2. Prior to injection, the PP pellets and composite mixtures were dried in an oven at 60 °C for 12 h. The thin injected plates were produced with a thickness of 1 mm and dimensions of 100 mm × 200 mm. The HIMACO LHS 150-80 injection machine was used for injecting the specimens at an injection pressure of 592 kgf·cm^−2^.

### 2.4. Characterization of Polypropylene and Composites

Dynamic mechanical thermal analysis (DMTA) was performed using a Q800 analyzer from TA Instruments, following the three-point bending method with two fixed points. The test specimens measured approximately 35 mm × 12 mm × 3 mm. A frequency of 1 Hz was applied, with a heating rate of 5 °C· min^−1^, over a temperature range of 28 to 100 °C.

Thermogravimetric analysis (TGA) was conducted using a NETZSCH TG 209F3 instrument. The analysis was performed in a nitrogen atmosphere with a flow rate of 50 mL min^−1^, over a temperature range of 25 to 800 °C, and a heating rate of 10 °C·min^−1^. An approximate sample mass of 10 g was used. To complement the analysis, the derivative thermogravimetric (DTG) curve was also obtained. Differential scanning calorimetry (DSC) analyses were performed using a NETZSCH DSC300 Classic system. An approximate sample mass of 10 g was used for all samples.

The DSC analysis was carried out under a nitrogen gas flow of 50 mL·min^−1^, within a temperature range of 25 to 200 °C, using a heating rate of 10 °C min^−1^ and two thermal cycles. The degree of crystallinity (Xc) was calculated using the following equation:(1)$$Xc=\frac{{∆H}_{melting}}{{∆H}_{m}^{0}\cdot (1-x)}\times 100$$
where ${∆H}_{melting}$ is the heat of fusion obtained from the DSC analysis (J g^−1^), ${∆H}_{m}^{0}$ is the heat of fusion for fully crystalline polypropylene (165 J·g^−1^), and x is the weight fraction of nanofiller in the sample [25,26].

The surface cross-section of the hybrid composites (cryogenically fractured specimens) was examined using a scanning electron microscope (SEM-JEOL JSM6060). All samples were sputtered with a layer of gold prior to SEM observation. The samples were oven-dried at 70 °C with air circulation for 24 h. The composites were then mounted on aluminum mounts using double-sided electrically conductive carbon adhesive tabs prior to analysis.

Mechanical tests were conducted using an EMIC universal testing machine, model DL 2000. Tensile tests were performed at a speed of 50 mm·min^−1^, while flexural tests were carried out at a speed of 1.5 mm·min^−1^, utilizing a 200 kgf load cell. Five specimens were tested for each sample, following the guidelines of ASTM D 638 for tensile tests and ASTM D 790 for flexural tests. Additionally, notched Izod impact tests were conducted in accordance with ASTM D 256. To assess the significance of the results, a one-way analysis of variance (ANOVA) was applied with a 95% confidence level (significance level of 0.05). The Tukey test was also performed to identify statistically significant differences between the composite samples and neat polypropylene. Statistical analyses were conducted using Jamovi software, version 2.3 [27,28] and included both tensile and flexural mechanical tests.

## 3. Results and Discussion

### 3.1. Dynamic Mechanical Thermal Analysis (DMTA)

Figure 3 shows the storage modulus (a), loss modulus (b), and tan delta (c) for neat polypropylene and the composites, obtained from DMTA. The storage modulus reflects the stiffness of the material, and as temperature increases, the value typically decreases due to enhanced mobility of the polymer chain segments within the matrix [28]. An increase in the storage modulus of the composites was observed when compared to neat PP. Highly oriented polymer chains behave like harmonic springs rather than shock absorbers, dissipating stress through viscous friction; hence, higher values are expected for crystalline polymers compared to amorphous ones [29] or for reinforced materials as composites, since the reinforcement acts by restricting the molecular mobility of the polymer chains. In this way, the incorporation of aramid fibers to the polymer matrix led to a significant increase in the storage modulus [30]. According to Jaiswal et al. (2021), aramid fibers enhance the energy absorption capacity of the composite material, which correlates with its ability to deform without incurring permanent structural damage [31].

When 0.1 wt.% GNPs was added to the composite containing PP and aramid fibers, a promising increase in the storage modulus was observed. However, at higher GNP concentrations (0.25 and 0.5 wt.%), a decrease in this property occurred, likely due to the restricted mobility of the polymer chains. This finding suggests that lower concentrations of GNPs are more effective than higher ones. At higher concentrations, increased graphene–graphene interactions may lead to agglomeration, which can deteriorate the interface between the graphene nanoplatelets and the polymer matrix. This phenomenon may negatively impact the mechanical properties of the interfacial layers [32]. The samples containing 2 wt.% MAPP also exhibited superior values compared to neat PP, although the results were still inferior to those of the composites containing GNPs (SAM7, SAM8, and SAM9). The chemical modification of polyolefins with MAPP aims to improve the compatibility and adhesion of PP to other materials due to the presence of polar carboxylic acid groups along the polymer chain [33]. This is supported by sample SAM9, which contained the highest concentration of graphene nanoplatelets (0.5 wt.%) and exhibited the best storage modulus result among all tested samples. These findings highlight the importance of modifying the polymer matrix to enhance the adhesion, compatibility, and dispersion of the added fillers in the composite.

The loss modulus, shown in Figure 3b, is associated with the amount of energy dissipated as heat during a cycle of periodic mechanical deformation. As observed, the loss modulus for neat PP was lower compared to the composites containing aramid fiber, MAPP, and GNPs. Additionally, for neat PP, the curve exhibited a linear decline with increasing temperature. In contrast, the composite materials displayed a more pronounced decrease only after 60 °C, indicating a strong interaction between the polymer matrix and aramid fibers. This interaction imposes constraints on the matrix, thereby enhancing stress transfer at the fiber interface [34]. The SAM9 sample, which has a higher GNP content, exhibited increased loss modulus. Due to their two-dimensional structure and highly specific surface area, GNPs interact more strongly with the surrounding polymeric chains. This enhanced interaction restricts molecular mobility during deformation, leading to greater dissipation as heat (higher loss modulus) [35].

The tan δ curves are shown in Figure 3c. All tested samples exhibited a similar trend, with an exponential increase up to 70 °C, followed by a stable pattern from this point until reaching 100 °C. The highest tan δ values were observed for neat PP, suggesting a more viscous material with lower elastic recovery capability [23,36]. In contrast, the incorporation of aramid fibers significantly enhanced the viscoelastic properties of the material. Notably, sample SAM9, containing graphene nanoplatelets, demonstrated the best elastic recovery capability, outperforming the other samples. This suggests that both aramid fibers and GNPs were effectively integrated into the polymer matrix, aided by the use of MAPP as a compatibilizer.

### 3.2. Thermogravimetric Analysis (TGA)

Figure 4 presents the thermogravimetric analysis (TGA and DTG) results for neat PP and composites containing MAPP, aramid fibers, and GNPs. The temperatures at 5% mass loss (T5%), 50% mass loss (T50%), the residue mass at 600 °C (Residue 600 °C), and the peak temperature of the DTG curves (TDTG1 and TDTG2) are summarized in Table 3.

At the onset of degradation (T5%), no significant difference was observed between the composite samples and neat PP. Typically, the addition of fillers like GNPs to the polymer matrix is expected to increase the degradation temperature, as demonstrated by Dallé et al. (2024) [37]. However, the low GNP concentration (up to 0.5 wt.%) may explain the similar T5% value among the tested samples. The largest increase in T5% was noted for sample SAM7 (424 °C), which contained MAPP (2 wt.%), aramid fiber (10 wt.%), and GNPs (0.1 wt.%). Additionally, a second degradation plateau was observed for the composite samples, occurring between 484 and 591 °C. The first plateau is attributed to the degradation of the PP matrix and MAPP, starting at 350 °C with the release of volatile organic compounds [26]. The second plateau corresponds to the degradation of aramid fibers, occurring at around 485 °C, consistent with the literature values [38].

DTG analysis revealed no significant differences between the samples; however, two characteristic peaks were identified. The first peak occurred between 455 and 464 °C, and the second between 581 and 585 °C. The presence of fillers in the polymer matrix did not shift the peak temperatures of the composites compared to neat PP. Sample SAM4, containing aramid fiber and a lower concentration of GNPs, exhibited a higher residual mass in the second plateau at 600 °C (Table 3). This result may be attributed to improved interaction between the polymer matrix and the lower GNP concentration, as well as the absence of MAPP.

### 3.3. Differential Scanning Calorimetry (DSC)

Figure 5 shows the DSC results for the first and second runs of neat PP and composite samples. The heat of melting, melting temperature, and crystallinity values are summarized in Table 4. The neat PP sample exhibited a more endothermic heat flow compared to the composites. With the addition of fillers, a decrease in endothermic flow is observed. Overall, the samples containing fillers showed a slight rightward peak shift, indicating a modest increase in melting temperature, with a maximum shift of 3 °C. This can be attributed to an increase in crystallinity (see Table 4), resulting in a more organized molecular structure and requiring more energy and time to melt. This behavior is consistent with the findings of Ye et al. (2024) [39], where the addition of small amounts of graphene oxide to a polypropylene–biochar mixture led to increased crystallinity. Sample SAM9, containing MAPP (2 wt.%), GNPs (0.5 wt.%), and aramid fibers (10 wt.%), exhibited the highest crystallinity (49.8%). In contrast, sample SAM6, without MAPP, showed lower crystallinity than SAM9. The use of a compatibilizing agent improves the interfacial adhesion between the filler and the polymer matrix but does not contribute to an increase in the crystallinity of the composite. It is worth noting that the crystallinity index is directly related to the crystalline structure of the polymer, i.e., amorphous polymers do not melt because they have no crystals, whereas crystalline polymers have a more organized structure (crystals) that melt with temperature. The higher the crystallinity index, the higher the melt temperature flows and, consequently, the higher the crystallinity index (compared to similar polymers). An increase in interfacial adhesion would only lead to an increase in the crystallinity index if the structure of the polymer were to change, i.e., to become more organized, and this does not appear to be the case, at least not according to these curves.

### 3.4. Scanning Electron Microscopy (SEM)

SEM images were examined to check for the possible presence of voids, fiber pull-out, agglomerates, etc. Figure 6A–R show SEM images of all the composites analyzed at two different magnifications. The images show that the incorporation of graphene or MAPP did not affect the morphology of the composites produced. Also, the presence of fiber pull-out or voids is not visualized, indicating that graphene nanoplatelets and MAPP have no influence on the morphology of the composites and consequently on the mechanical properties (presented in the next section). Graphene nanoplatelets and/or MAPP appear to maintain the morphological structure of the composite and may improve biological or cytological performance (not tested here).

### 3.5. Mechanical Properties

Figure 7 shows the results of tensile strength at break, elongation at break, and Young’s modulus for the composite studied. The tensile strength (Figure 7a) of neat polypropylene was lower than that of composites containing aramid fibers, MAPP, and GNPs. The typical tensile strength of PP ranges from 31.0 to 41.4 MPa [1]; however, the neat PP in this study exhibited a value below the reference range (12.8 MPa). In contrast, aramid fibers typically exhibit tensile strengths around 3000 MPa [40]. Additionally, graphene, one of the strongest materials known, can achieve values up to 130 GPa [41]. With the addition of aramid fiber and GNPs to the polymer matrix, the tensile strength of the composite (SAM8-ITP) reached 46.8 MPa, representing a 265% increase in this property. The presence of both materials acts as reinforcement, enhancing stiffness. Due to the higher energy absorption and impact properties of aramid fibers and GNPs, the composite exhibits improved robustness. Figure 7b presents the elongation at break results for all samples. Neat polypropylene exhibited the highest value (281.4%) compared to the composites, which is consistent with the literature [1].

The composites containing aramid fibers and GNPs exhibited elongation at break values ranging from 16.1% to 18.8%. This reduction is due to the inherently lower elongation at the break of the aramid fiber (1.5 to 3.6%) and graphene (up to 30%), both of which contribute significantly to the mechanical properties of the composite.

When the Young’s modulus, shown in Figure 7c, was observed, the SAM3-ITP (302.1 MPa) and SAM9-ITP (302.6 MPa) samples exhibited lower values compared to neat PP (536.4 MPa), which may indicate that the absence or a higher quantity of GNPs could affect the stiffness of the composite. On the other hand, the SAM5-ITP and SAM8-ITP samples were the most promising, with values of 798.1 MPa and 824.1 MPa, respectively. The slight increase in the elastic modulus of the SAM8-ITP sample compared to SAM5-ITP is attributed to MAPP, which enhances the compatibility and adhesion of the polymer matrix with the filler materials. Other studies also demonstrate that there is an improvement in the compatibility and adhesion of the polymer matrix with graphene oxide when a compatibilizer such as MAPP is used [42,43].

In general, the mechanical properties were more drastically affected by the aramid fiber than by the graphene nanoplatelets and/or MAPP. This indicates that the aramid fiber plays a major role in the mechanical properties tested and that the interface (where the stress is transferred) is not significantly improved by the incorporation of graphene or MAPP.

The ANOVA results for the tensile and flexural mechanical tests indicated statistical differences between the samples. However, one-way ANOVA alone does not specify which specific samples differ. To identify these differences, Tukey’s test was performed. All composite samples exhibited statistically significant differences from neat polypropylene, with *p*-values < 0.001, confirming a notable modification of the polymer matrix.

In the tensile strength analysis, significant differences (*p* < 0.05) were found only between the SAM4-ITP and SAM6-ITP samples, as well as between the SAM6-ITP and SAM8-ITP samples. Regarding elongation at break, no statistical differences were observed among the composites. For Young’s modulus, the SAM3-ITP and SAM9-ITP samples showed statistically significant differences when compared to all other samples. As shown in Figure 6C, these samples exhibited a reduction in Young’s modulus relative to the other composites and neat polypropylene.

The flexural test results are presented in Figure 8. The flexural strength (Figure 8a) reflects the material’s capacity to resist deformation under applied load, a critical property for high-performance polymers and composites [44].

In the present study, an improvement in the performance of the composites was observed when compared to neat PP. The SAM5-ITP sample showed the best result among the others, reaching an average flexural strength of 62.4 MPa, while the neat PP showed a value of 36.73 MPa, consistent with the reference data. Additionally, the SAM6-ITP sample exhibited a flexural strength of 59.1 MPa. However, samples containing MAPP did not demonstrate a significant enhancement in this property, although they were still superior to neat PP. Similarly, Arı et al. [14] reported that the incorporation of aramid fiber improves the flexural strength of polypropylene-based composites. When comparing the SAM2-ITP sample with the SAM5-IPT sample, the presence of GNPs appears to contribute to an increase of up to 8.5 MPa in flexural strength.

On the other hand, as shown in Figure 8b, the composites exhibited a slight reduction in flexural elongation compared to neat PP. This result is notable because, although neat polypropylene already has low flexural elongation, the composites did not decrease to the extent that they become more brittle. Modified materials can often exhibit a considerable decrease in flexural elongation, which can make them less ductile for certain applications, such as materials for bulletproof vests [37,45].

The flexural modulus results for the composites were higher than those of neat polypropylene, as shown in Figure 8c. The SAM5-ITP and SAM6-ITP samples showed values of 2206.3 MPa and 2235.5 MPa, respectively, while neat polypropylene reached only 1193.9 MPa [46], which corresponds to an increase of 1.84 and 1.87 times for the SAM5-ITP and SAM6-ITP, respectively. These findings indicate that a controlled addition of GNPs (0.25 wt.%) effectively enhances the flexural properties of the composite. Furthermore, while the inclusion of MAPP did not further improve this property, the modified composites still outperformed neat polypropylene.

The flexural tests revealed that the incorporation of AF, MAPP, and GNPs into the polymer matrix caused significant structural changes in the composites compared to neat PP, as indicated by the *p*-value (<0.001). However, the statistical analysis of flexural elongation (Figure 8b) showed no significant differences among the composite samples, except for the SAM9-ITP, which was distinct from both SAM3-ITP and SAM7-ITP. Additionally, only the SAM3-ITP, SAM7-ITP, and SAM8-ITP samples exhibited statistical differences when compared to neat PP. In terms of flexural modulus (Figure 8c), most composite samples did not show statistically significant differences. The exceptions were SAM6-ITP ≠ SAM7-ITP and SAM9-ITP ≠ both SAM5-ITP and SAM6-ITP.

The impact resistance study, shown in Figure 9, demonstrates a significant enhancement in the performance of the composites compared to neat PP. The impact resistance of neat PP was 2.85 ± 1.19 kJ·m^2^, while the sample with aramid fibers achieved a value of 6.44 ± 0.81 kJ·m^2^, indicating a substantial improvement. This enhancement can be attributed to the superior absorption capacity of aramid fibers relative to polypropylene [47]. The inclusion of MAPP as a compatibilizer agent further increased the impact resistance, reaching 8.94 ± 1.68 kJ m^2^, suggesting improved interfacial adhesion between the aramid fiber and the polymer matrix. Conversely, the addition of GNPs led to a reduction in impact resistance, although the values remained higher than that of neat PP. Statistical analysis using Tukey test revealed no significant differences between the composite samples containing GNPs, regardless of the presence of MAPP.

## 4. Conclusions

In this study, polypropylene composites reinforced with aramid fibers (AF) and graphene nanoplatelets (GNPs) were produced and evaluated for their mechanical properties. The results indicated an increase in the mechanical properties of the composites containing AF and GNPs. In the dynamic mechanical analysis (DMA) results, both the storage modulus and the loss modulus were higher for all modified samples compared to the neat polymer matrix. The addition of 0.1 wt.% GNPs to the composite containing polypropylene (PP) and aramid fibers (AF) resulted in a promising increase in the storage modulus. However, this effect diminished at concentrations of 0.25 and 0.5 wt.%. The samples containing 2 wt.% MAPP also exhibited superior results compared to neat polypropylene, although they were inferior to the composites with GNPs. The SAM8-ITP sample, containing AF and GNPs, exhibited the highest tensile strength at a break of 46.8 MPa, representing an increase of up to 3.6 times compared to the neat PP matrix. Additionally, due to the low elongation at a break of aramid fibers, a significant reduction in this property was observed in the composites containing AF, GNPs, and MAPP. The Young’s modulus was lower both in the absence of GNPs and at its highest concentration, suggesting that the amount of graphene affects the material’s stiffness. The flexural moduli of the composites were also higher compared to the neat polymer matrix. The samples containing 0.25 and 0.5 wt.% GNPs, without MAPP, exhibited the highest flexural moduli values of 2206.3 and 2235.5 MPa, respectively. In comparison, the flexural modulus of neat PP was 1193.9 MPa. With promising results, the main contribution of this study is the investigation of the mechanical and dynamic mechanical properties of composites using an emerging approach: the combination of fibers as reinforcement with graphene nanoplatelets, the latter being considered a new milestone in the development of high-performance materials. For future studies, it is suggested to discuss in depth the failure mechanism and to establish theoretical formulas. Furthermore, the injected thin plates could be used in future studies to improve the mechanical and dynamic resistance in ballistic applications, replacing the traditional films already studied by injected thin plates containing aramid and graphene nanoplatelets. In addition, the injection method used to produce the thin sheets allows the possibility of reusing the aramid fibers used in ballistic components. Despite the promising prospects of graphene in various fields such as dentistry, there are still several challenges that need to be overcome before these materials can be fully commercialized. For example, it is important to study the structure–property relationship of this type of material.

## Figures and Tables

**Figure 1 polymers-17-00374-f001:**
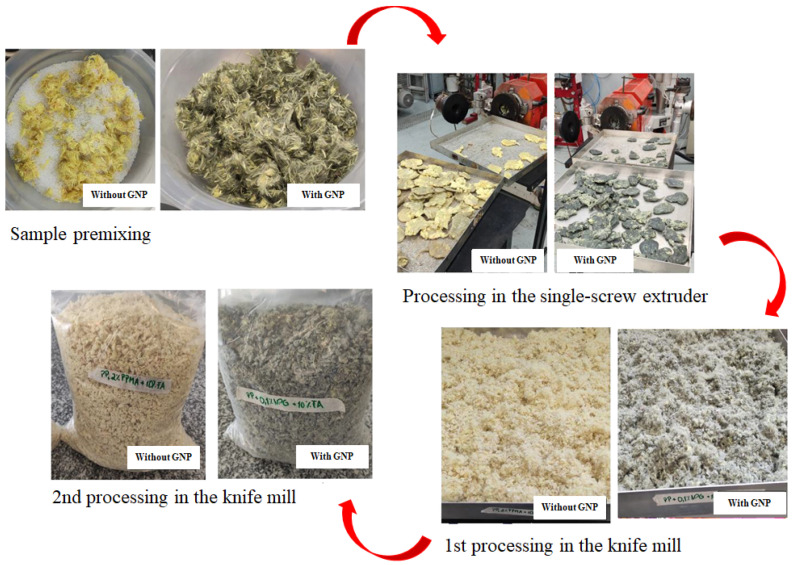
Stages of processing in the single-screw extruder and knife mill for the samples studied.

**Figure 2 polymers-17-00374-f002:**
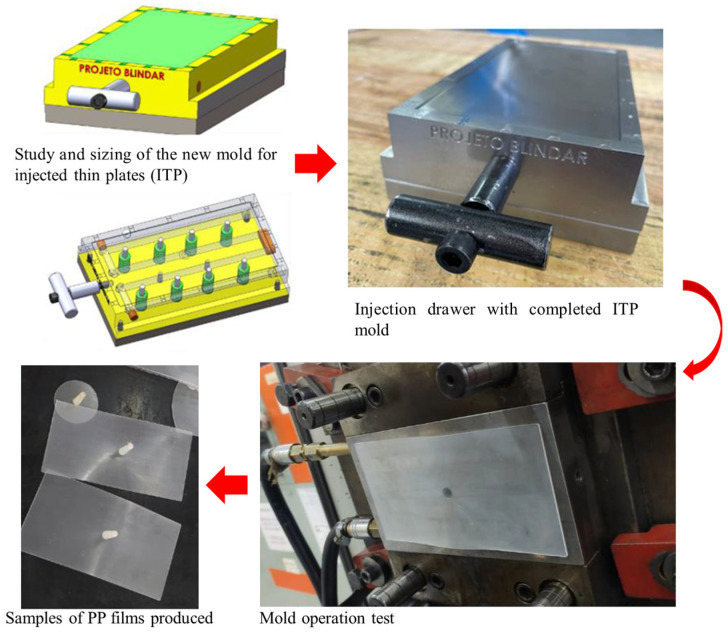
Development of the mold for thin-plate injection.

**Figure 3 polymers-17-00374-f003:**
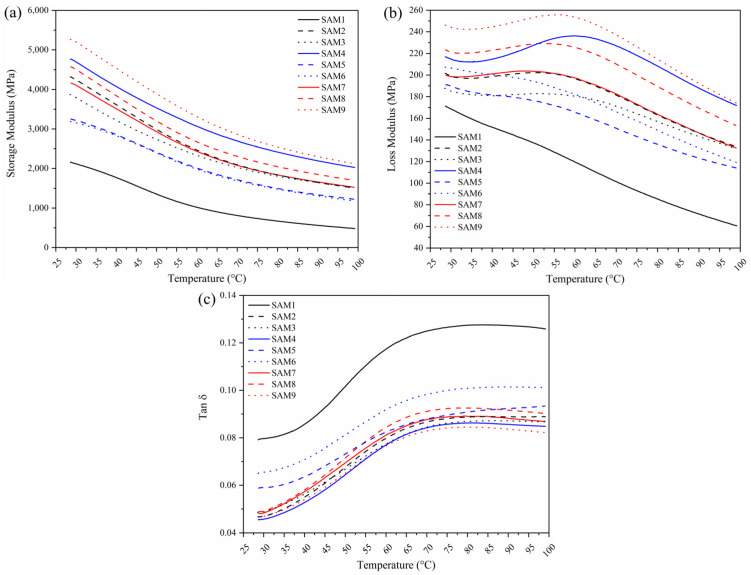
Storage modulus (**a**), loss modulus (**b**), and tan δ (**c**) of neat polypropylene and composite samples containing aramid fibers, graphene nanoplates (GNPs), and MAPP.

**Figure 4 polymers-17-00374-f004:**
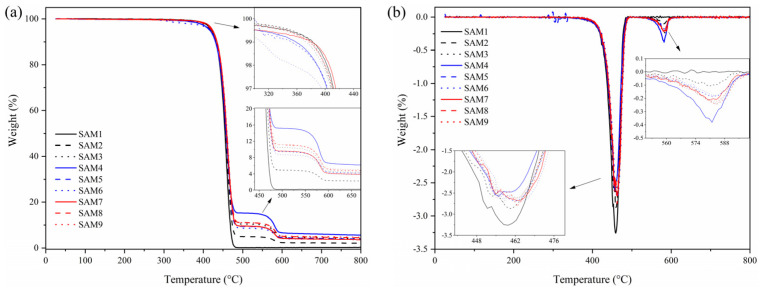
(**a**) TGA and (**b**) DTG analyses of the composites compared to the neat PP matrix.

**Figure 5 polymers-17-00374-f005:**
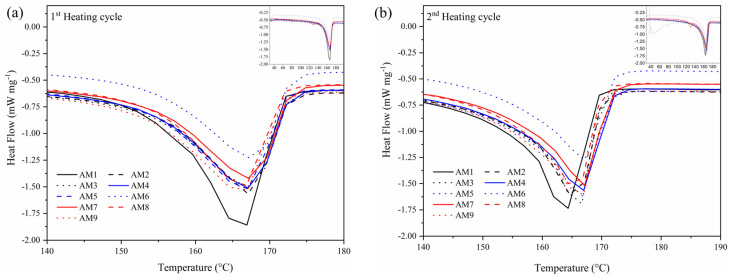
DSC analysis of (**a**) first heating cycle and (**b**) second heating cycle for the neat PP matrix and composite samples.

**Figure 6 polymers-17-00374-f006:**
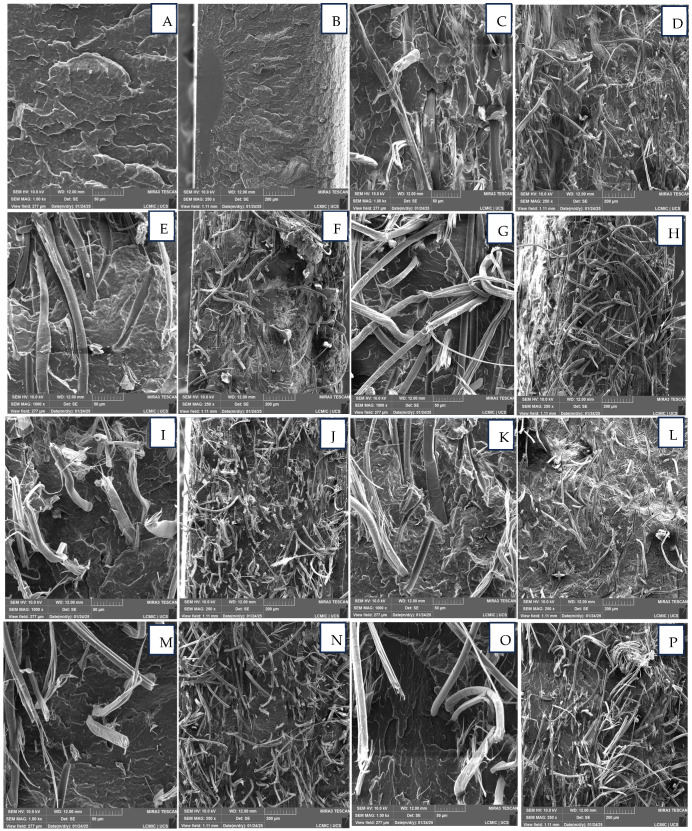
Scanning electron microscopy (SEM) of composites and neat matrix at two different magnifications indicated in the images: 50 µm and 200 µm. (**A**,**B**) AM1, (**C**,**D**) AM2, (**E**,**F**) AM3, (**G**,**H**) AM4, (**I**,**J**) AM5, (**K**,**L**) AM6, (**M**,**N**) AM7, (**O**,**P**) AM8, and (**Q**,**R**) AM9.

**Figure 7 polymers-17-00374-f007:**
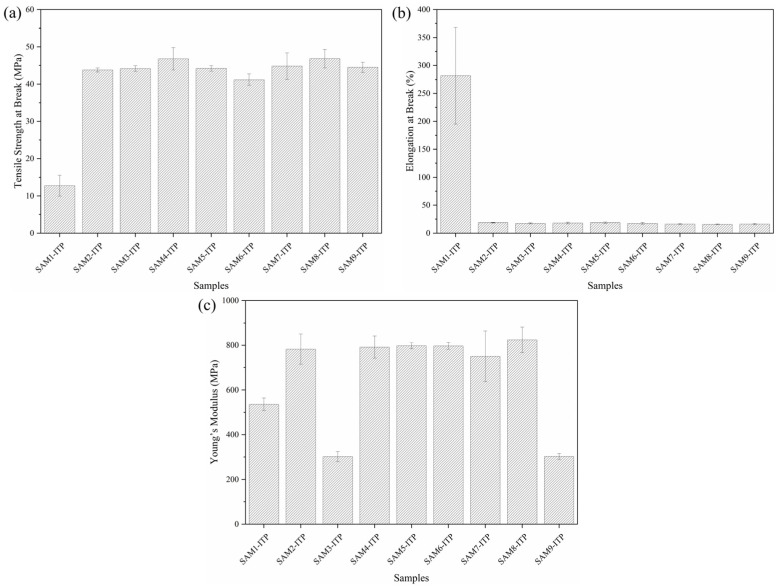
Tensile strength at break (**a**), elongation at break (**b**), and Young’s modulus (**c**) of composites samples manufactured via the thin-plate injection process.

**Figure 8 polymers-17-00374-f008:**
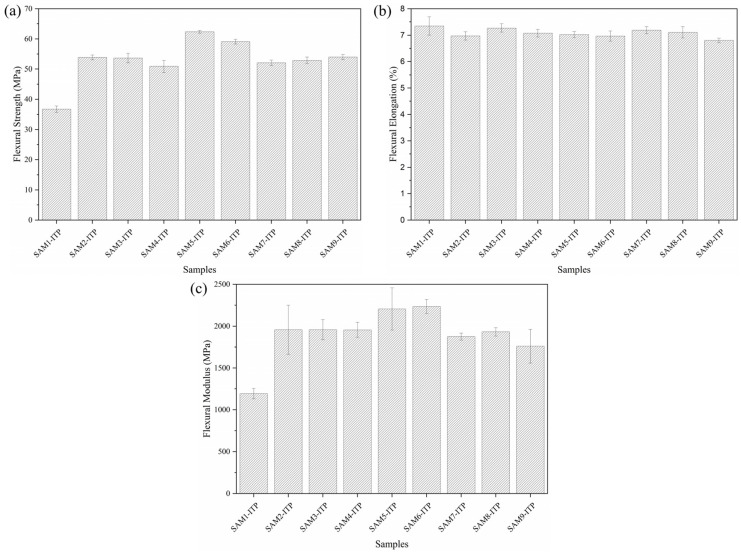
Flexural resistance (**a**), flexural elongation (**b**), and flexural modulus (**c**) for composite samples produced via the thin-plate injection process.

**Figure 9 polymers-17-00374-f009:**
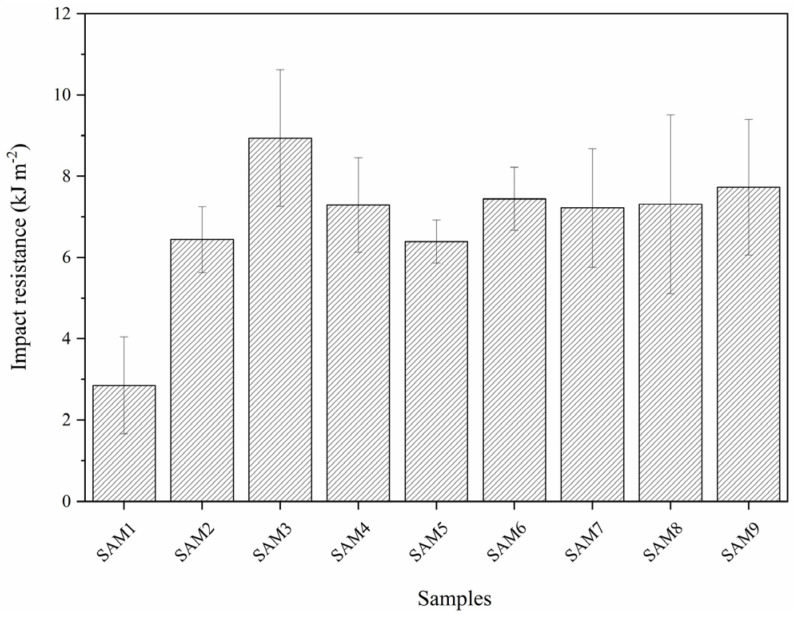
Impact resistance analysis of composites and neat matrix.

**Table 1 polymers-17-00374-t001:** Nomenclature and composition of the samples prepared for this study.

			Compositions		
Samples	Polypropylene (wt.%)	Aramide Fiber (wt.%)	Fiber Length (mm)	MAPP (wt.%)	Graphene Nanoplatelets (wt.%)
SAM1	100	0	0	0	0
SAM2	90	10	10	0	0
SAM3	88	10	10	2	0
SAM4	89.9	10	10	0	0.1
SAM5	89.75	10	10	0	0.25
SAM6	89.5	10	10	0	0.5
SAM7	87.9	10	10	2	0.1
SAM8	87.75	10	10	2	0.25
SAM9	87.5	10	10	2	0.5

**Table 2 polymers-17-00374-t002:** Injection parameters of neat PP thin plates.

Parameters	Values Used
Injection time	4 s
Cooling time	20 s
Injection pressure	534.8 kgf cm^−2^ (28% of the equipment)
Injection flow	42.3 cm³ s^−1^ (45% of the equipment)
Temperature profile	160 °C, 175 °C e 190 °C
Nozzle heating	50% (10 s ON–10 s OFF)
Mold temperature	40 °C

**Table 3 polymers-17-00374-t003:** Thermal properties of neat PP and composite samples evaluated by TGA and DTG (obtained from Figure 4).

	TGA			DTG	
Sample	T_5%_ (°C)	T_50%_ (°C)	Residue_600°C_ (wt.%)	T_DTG1_ (°C)	T_DTG2_ (°C)
SAM1	421	455	0.19	459	-
SAM2	420	457	2.36	460	581
SAM3	420	459	4.68	458	584
SAM4	417	458	6.63	456	582
SAM5	418	458	4.48	462	585
SAM6	417	460	4.10	464	584
SAM7	424	459	4.18	464	581
SAM8	422	460	5.09	463	584
SAM9	422	458	5.30	455	584

**Table 4 polymers-17-00374-t004:** Thermal properties from DSC analysis (Figure 5b) of neat PP and composite samples.

Sample	Heat Melting (J g^−1^)	Crystallinity (%)	Melting Temperature (°C)
AM1	90.2	54.7	164
AM2	85.4	57.5	165
AM3	86.4	59.5	167
AM4	86.1	58.0	167
AM5	85.1	57.5	167
AM6	83.1	56.3	167
AM7	88.3	60.9	167
AM8	84.3	58.2	165
AM9	91.1	63.1	167

## Data Availability

Data are contained within the article.

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
