# Peer review of "Exploring the Synergistic Effect of Short Aramid Fibers and Graphene Nanoplatelets on the Mechanical and Dynamic Mechanical Properties of Polypropylene Composites Prepared via Thin-Plate Injection"

_polymers, 2025, doi:10.3390/polym17030374_

Round 1

Reviewer 1 Report

Comments and Suggestions for Authors

The manuscript concerns composites with fillers that interact in interesting ways. Specifically, it investigates composites with a combination of aramid fibers, graphene, and maleic anhydride grafted polypropylene. The manuscript should be of interest to the readership, is well-written, and is well-organized. I have some optional comments that the authors may consider as part of a minor revision.

1. pg 6, "The storage modulus ... typically decreases with increasing temperature due to the increased mobility of the polymer chains within the matrix". The authors should consider this statement carefully. It does not seem correct to me. Polymers generally become stiffer with increasing temperature because their elasticity is due to entropy. "Increased mobility" is not entirely clear to me, but if it means increased entropy then the polymers should be stiffer.

2. Similar comment: there is a claim that aramid fibers increase the storage modulus by restricting the mobility of the polymer chains. Does it need to be due to chain mobility? Can it not just be due to a higher volume fraction of a stiffer material?

3. Similar comment regarding chain mobility and graphene nanoplatelets. Perhaps the authors can either revise their claims or better explain why they believe polymer mobility relates to storage modulus in this way.

Author Response

ID polymers-3434239

Status – Pending major revisions

Article type – article

Title Investigation of the synergistic effect of short aramid fibers and graphene nanoplatelets on the mechanical and dynamic mechanical properties of polypropylene composites prepared by thin-plate injection moulding.

Reviewer#1

Comments and Suggestions for Authors

The manuscript concerns composites with fillers that interact in interesting ways. Specifically, it investigates composites with a combination of aramid fibers, graphene, and maleic anhydride grafted polypropylene. The manuscript should be of interest to the readership, is well-written, and is well-organized. I have some optional comments that the authors may consider as part of a minor revision.

Answer: We appreciate the effort and considerations of the reviewer. We tried our best to address all comments and incorporate in the manuscript.

  1. pg 6, "The storage modulus ... typically decreases with increasing temperature due to the increased mobility of the polymer chains within the matrix". The authors should consider this statement carefully. It does not seem correct to me. Polymers generally become stiffer with increasing temperature because their elasticity is due to entropy. "Increased mobility" is not entirely clear to me, but if it means increased entropy then the polymers should be stiffer.

Answer: Thank you for your comment. Stiffer materials should present higher storage modulus than softer ones. This is represented by higher storage modulus values. All materials, including polymeric ones, have a specific atomic disposal (which reflects on their properties as well their conformational and configurational states, resulting in amorphous or crystalline materials, for example) Also, all materials (with some exceptions as zirconia (ceramic)) with temperature storage more energy as heat and consequently the polymeric chains (in the case of polymers) apart each other aiming to dissipate the energy received. This phenomenon is called reptation (similar to snakes). Reptation is possible due to an increase in the internal energy causing an increase in the free volume of the polymer due to a higher number of conformational states which the polymer can achieve for the same energy level. Higher free volume causes a decrease of the capacity of the polymer on storage energy; hence the storage modulus decreases with temperature. This is the reason why the storage modulus decreases as it passes through the glass transition temperature, where an abrupt decrease of the modulus is observed. In this state, the distribution of energy is more “equal”, which would result in a higher entropy, but not becomes the material stiffer. One exception for polymeric materials is some elastomers which, by mechanical stress can increase the modulus after deformation by an increase in the internal entropy – but in this case, we cannot consider the temperature effect.

  1. Similar comment: there is a claim that aramid fibers increase the storage modulus by restricting the mobility of the polymer chains. Does it need to be due to chain mobility? Can it not just be due to a higher volume fraction of a stiffer material?

Answer: The reviewer is right. Stiffer materials promote higher storage modulus, but it is important to mention that the response is primarily of the polymer and not the filler. There are many examples in literature in which glass fiber is incorporated, for example on a softer polymeric matrix, and the values decrease. The main point is to know if the filler somehow interferes or not in the polymer chain mobility. That is the reason why the interface plays a major role on composite materials. All external energy received by the composite is primarily received by the polymer and transferred for the fibers via interface. If the interface is poor, a detachment of the fiber occurs and the storage modulus decreases with temperature. If the interface is good, the energy is transferred by the polymer to the fiber (which has a pure elastic behavior) which retains most of the external stress and decreasing the polymeric chains mobility (considering the whole polymeric chains). When this stress increases, the polymeric chains (which could stay “still”) start to increase their mobility aiming to dissipate the internal energy, and a similar process as mentioned in the former question is observed. Hence, a higher volume fraction increases the storage modulus due to be a stiffer material (as noted by the reviewer) and due to a decrease in the molecular mobility of the polymeric chains. Again, since the analysis is performed on polymers it is easier to think about how the added components can interfere in this behavior (comparing the neat polymer).

  1. Similar comment regarding chain mobility and graphene nanoplatelets. Perhaps the authors can either revise their claims or better explain why they believe polymer mobility relates to storage modulus in this way.

Answer: Same explanation as above.

Reviewer 2 Report

Comments and Suggestions for Authors

The research methods are well-known. The research content is interesting, but only testing and analysis were conducted without further discussion, such as failure mechanism and morphology analysis, establishment of theoretical formulas, and so on. Given this, it is strongly recommended to include current shortcomings and future prospects in the conclusion.

Author Response

ID polymers-3434239

Status – Pending major revisions

Article type – article

Title Investigation of the synergistic effect of short aramid fibers and graphene nanoplatelets on the mechanical and dynamic mechanical properties of polypropylene composites prepared by thin-plate injection moulding.

Reviewer#2

Comments and Suggestions for Authors

The research methods are well-known. The research content is interesting, but only testing and analysis were conducted without further discussion, such as failure mechanism and morphology analysis, establishment of theoretical formulas, and so on. Given this, it is strongly recommended to include current shortcomings and future prospects in the conclusion.

Answer: We appreciate the effort and consideration of the reviewer. We tried our best to address all comments and incorporate into the manuscript.  It was included optical microscopy to evaluate the microstructure and failure surface of the samples, but further discussion such as theoretical formulas and so on, are not in the scope of the paper. For this reason, as suggested by the reviewer, it was highlighted the shortcomings and future prospects in the conclusion.

Reviewer 3 Report

Comments and Suggestions for Authors

Please find below my comments on the manuscript titledInvestigation of the synergistic effect of short aramid fibres and graphene nanoplatelets on the mechanical and dynamic mechanical properties of polypropylene composites prepared by thin-plate injection moulding”:

1.     Page 9 line 278, “With the addition of fillers, a shift towards exothermic flows was observed.” I don’t think this sentence is correct. It is still endothermic flow but decreases.

2.     Page 9 line 279. , “Overall, the samples containing fillers showed a slight peak shift to the right, indicating a slight increase in melting temperature, with a maximum shift of 3 °C.” The authors need to discuss the reason for the increase in melting temperature.

3.     Page 9, line 285, “The use of a compatibilizer improves the interfacial adhesion between the filler and the polymer matrix, contributing to the increased 286 crystallinity in the composite.” What is the relation between interfacial adhesion to the crystallinity? .In general, the authors need to discuss the DSC results more appropriately.

4.     The authors need to explain more about the decrease of elongation at break. The elongation of the composite is higher than AF, is it fiber fracture or fiber pull-out that occurred the AF?

5.     The authors need to include the microstructure and failure surface of the samples.

6.     The tensile test result is inconsistent. The addition of GNP and MAPP didn’t show much difference. Need to explain more about the results.

7.     The authors need to observe the presence of agglomerations and voids in the composites.

Author Response

ID polymers-3434239

Status – Pending major revisions

Article type – article

Title Investigation of the synergistic effect of short aramid fibers and graphene nanoplatelets on the mechanical and dynamic mechanical properties of polypropylene composites prepared by thin-plate injection moulding.

Reviewer#3

Please find below my comments on the manuscript titled “Investigation of the synergistic effect of short aramid fibres and graphene nanoplatelets on the mechanical and dynamic mechanical properties of polypropylene composites prepared by thin-plate injection moulding”:

Answer: We appreciate the effort and considerations of the reviewer. We tried our best to address all comments and incorporate in the manuscript.

  1. Page 9 line 278, “With the addition of fillers, a shift towards exothermic flows was observed.” I don’t think this sentence is correct. It is still endothermic flow but decreases.

Answer: The reviewer is right. We correct the sentence.

  1. Page 9 line 279. , “Overall, the samples containing fillers showed a slight peak shift to the right, indicating a slight increase in melting temperature, with a maximum shift of 3 °C.” The authors need to discuss the reason for the increase in melting temperature.

Answer: A sentence was included about the increase observed. Basically, is due to the increase in the crystallinity index (Table 4) which generates a more organized molecular system, requiring more time/temperature to melt the crystals.

  1. Page 9, line 285, “The use of a compatibilizer improves the interfacial adhesion between the filler and the polymer matrix, contributing to the increased 286 crystallinity in the composite.” What is the relation between interfacial adhesion to the crystallinity? In general, the authors need to discuss the DSC results more appropriately.

Answer: The reviewer is right. We got two issues here: 1) the crystallinity increases Tm and 2) a better interfacial adhesion is not directly connected with it. A stronger interface can increase Tm only by restricting the molecular mobility of the polymeric chains and requiring more energy to melt the crystals. Observing again Table 4, no significant difference was observed by incorporation of the compatibilizer agent. We rewrite the sentence.

  1. The authors need to explain more about the decrease of elongation at break. The elongation of the composite is higher than AF, is it fiber fracture or fiber pull-out that occurred the AF?

Answer: It was included in the manuscript a deeper explanation about this behavior. Scanning electron microscopy helps us to explain this behavior, as suggested by the author below. The elongation at break is higher for the neat resin because the incorporation of fiber let the material stiffer. It was highlighted in the manuscript.

  1. The authors need to include the microstructure and failure surface of the samples.

Answer: It was included scanning electron microscopy to evaluate the microstructure and failure surface of the samples.

  1. The tensile test result is inconsistent. The addition of GNP and MAPP didn’t show much difference. Need to explain more about the results.

Answer: SEM results help us to discuss the mechanical tests. According to the results presented in this study, it is noted that aramid fiber that plays a major role in the mechanical properties. MAPP and graphene nanoplatelets did not seems to strongly influence the mechanical properties and/or modify the fiber/matrix interface. The main conclusion is that this indifference can be positive seems, since the morphological structure is not affected by graphene or MAPP, maintaining the mechanical properties and potentially having a positive effect on the biological or cytological properties of the composites (not tested but verified on literature).

  1. The authors need to observe the presence of agglomerations and voids in the composites.

Answer: The presence of agglomerations and voids was analyzed by optical microscopy.

Round 2

Reviewer 3 Report

Comments and Suggestions for Authors

Accept in present form